# Circulating cell-free RNA in blood as a host response biomarker for detection of tuberculosis

Adrienne Chang [1,9], Conor J. Loy[1,9], Daniel Eweis-LaBolle [1,9], Joan S. Lenz[1], Amy Steadman[2], Alfred Andrama[3], Nguyen Viet Nhung[4], Charles Yu[5], William Worodria[3], Claudia M. Denkinger [6], Payam Nahid[7], Adithya Cattamanchi [7,8] & Iwijn De Vlaminck [1] ✉

Tuberculosis (TB) remains a leading cause of death from an infectious disease worldwide, partly due to a lack of effective strategies to screen and triage individuals with potential TB. Whole blood RNA signatures have been tested as biomarkers for TB, but have failed to meet the World Health Organization's (WHO) optimal target product profiles (TPP). Here, we use RNA sequencing and machine-learning to investigate the utility of plasma cell-free RNA (cfRNA) as a host-response biomarker for TB in cohorts from Uganda, Vietnam and Philippines. We report a 6-gene cfRNA signature, which differentiates TB-positive and TB-negative individuals with AUC = 0.95, 0.92, and 0.95 in test, training and validation, respectively. This signature meets WHO TPPs (sensitivity: 97.1% [95% CI: 80.9-100%], specificity: 85.2% [95% CI: 72.4-100%]) regardless of geographic location, sample collection method and HIV status. Overall, our results identify plasma cfRNA as a promising host response biomarker to diagnose TB.

The outcome of infection after exposure to *Mycobacterium tuberculosis*, the causative agent of tuberculosis (TB), can vary greatly and is determined by a complex interplay between host immunity and bacterial persistence[1]. Current diagnostic tests for TB are not sensitive to the full spectrum of disease states, are unable to determine if an infection has been cleared, cannot distinguish between latent, incipient, and subclinical disease, and cannot predict progression to active TB[2,3]. This lack of sensitivity and specificity in diagnostic tests presents a major challenge in the management and control of TB.

Transcriptomic signatures, or changes in host-cell gene expression, can provide valuable insight into the host response to TB. Yet, while several whole blood RNA (wbRNA) signatures have been identified, none have met the optimal target product profiles (TPPs)

recommended by the World Health Organization (WHO) for a non-sputum-based triage or diagnostic test[4-6]. In this study, we investigated plasma cell-free RNA (cfRNA) as a potential new class of host biomarkers for TB. cfRNA is an analyte that can provide information beyond what can be achieved with conventional wbRNA signatures for two key reasons. First, cfRNA is released by cells in both the blood compartment and from vascularized solid tissues and thus contains information about systemic immune dynamics and immune-tissue interactions. Second, cfRNA is released predominantly from dead or damaged cells as opposed to wbRNA, which originates from a mix of live and dead cells. Thus, cfRNA may provide far clearer insights into pathways of cell death and tissue injury that would not be available with wbRNA profiling.

[1]Meinig School of Biomedical Engineering, Cornell University, Ithaca, NY, USA. [2]Global Health Labs, Inc., Bellevue, WA, USA. [3]World Alliance for Lung and Intensive Care Medicine in Uganda, Kampala, Uganda. [4]National Lung Hospital, Hanoi, Vietnam. [5]De La Salle Medical and Health Sciences Institute, Dasmarinas, Philippines. [6]University Hospital Heidelberg & German Center of Infection Research, Heidelberg, Germany. [7]UCSF Center for Tuberculosis, University of California San Francisco, San Francisco, CA, USA. [8]Division of Pulmonary and Critical Care Medicine, University of California Irvine, Orange, CA, USA. [9]These authors contributed equally: Adrienne Chang, Conor J. Loy, Daniel Eweis-LaBolle. ✉e-mail: vlaminck@cornell.edu

To evaluate the performance of plasma cfRNA as a host signature of TB, we conducted unbiased profiling of plasma cfRNA in patient cohorts spanning three countries. Our analysis identified a 6-gene signature with accuracy of 91.8%, which exceeds the optimal WHO TPPs for a triage test. Comparison to matched whole blood samples highlighted the differences in the origin of plasma cfRNA and whole blood RNA. These results lay the groundwork for the development and clinical validation of a cfRNA host response test for the detection of TB.

## Results

### Plasma cell-free RNA signatures of tuberculosis

We performed a case-control study to investigate the cfRNA signatures of tuberculosis. We analyzed plasma samples from a total of 251 individuals with a cough lasting at least two weeks who were enrolled in three different cohorts (Cohort 1, Cohort 2, and Cohort 3) at outpatient clinics in Uganda, Vietnam, and the Philippines (Table 1 and Fig. 1A). Individuals included in the "TB positive" group were required to have 1) a positive Xpert MTB/RIF Ultra on sputum, urine, or contaminated Mycobacterial Growth Indicator Tube (MGIT) specimen; 2) a positive sputum MGIT or solid culture; or, 3) two trace Xpert Ultra results on sputum or contaminated MGIT. All other individuals ("TB negative" group) had at least one negative Xpert Ultra result, two negative cultures in MGIT or solid media, and repeated negative sputum tests and/ or clinical improvement without TB treatment at two to three months follow-up. Of the 251 plasma samples analyzed in our study, 142 (56.6%) were from individuals diagnosed with microbiologically confirmed TB, and 37 (14.7%) were from individuals living with HIV. We used next-generation sequencing to profile the RNA in these blood samples. This yielded a mean of 31,737,786 reads per sample in plasma (range: 7,200,555 to 64,206,141 reads); Supplementary Fig. 1).

To determine the cell types-of-origin of the cfRNA in our samples, we used a reference-based deconvolution algorithm (BayesPrism). This algorithm uses a reference dataset, in this case the Tabula Sapiens single-cell RNA-seq atlas, and bayesian inference to identify the cell types from which the cfRNA in our samples originated[7–10]. We found that the plasma cfRNA in individuals with respiratory symptoms was primarily derived from platelets, myeloid progenitor cells, B cells, endothelial cells, natural killer cells, monocytes, and neutrophils (Fig. 1B). We observed batch effects in the platelet contribution. There was a significant difference in the platelet contribution between all three cohorts (mean fraction: 0.11, 0.61, 0.36 for Cohort 1, Cohort 2, and Cohort 3, respectively). These differences were primarily due to the different sample collection tubes and plasma collection procedures used by the Cohort 1, Cohort 2, and Cohort 3 studies (Methods).

To gain insight into the host response to TB and evaluate if cfRNA is informative of TB status, we performed differential abundance analysis on all 251 samples (DESeq2[11], Methods). Using a Benjamini-Hochberg adjusted $p$-value cutoff of <0.05, we identified 1956 differentially abundant genes between TB positive and negative groups (Fig. 1C and Supplementary Data 1). TB was associated with elevated levels of macrophage markers (*MARCO*, *SOCS3*, *FCGR1A*, *MPO*, *C1QB*), neutrophil markers (*ELANE*, *FCGR3A*, *S100A8*, *S100A9*, *ERG*), interferon genes (*IFI27L1*, *IFIT2*, *IFIT3*, *IFITM3*, *IRF1*), and antimicrobial genes

(*AZU1*, *CTSG*, *DEFA4*, *STAT1*, *GBP1*, *GBP2*, *GBP4*, *GBP5*, *GBP6*) compared to the TB negative group. We also observed elevated levels of lung-specific markers (*SFTPC*, *SFTPB*, *SLC34A2*, *SFTPA1*, *SFTPA2*) in individuals with TB, providing insight into ongoing pathogenic processes[12,13]. To confirm the relevance of these findings, we performed Ingenuity Pathway Analysis (Fig. 1D; Supplementary Data 2). The top canonical pathways enriched in TB (filtered by |-log($p$-value)|>1.0 and |z-score| >2.55) included neutrophil degranulation, interferon gamma signaling, and antimicrobial peptide pathways. We observed similar enriched pathways using gene set enrichment of KEGG, GO and Hallmark pathways (Supplementary Data 3).

### Evaluating machine learning algorithms for gene-expression-based TB diagnostics

The results from the differential expression analysis, including enrichment of clinically relevant pathways and satisfactory separation of sample groups using correlation-based clustering, suggested that differentially abundant genes in plasma cfRNA could be used to develop triage tests and diagnostic assays for TB. To test this, we split the samples into a "discovery" cohort (Cohort 1 and Cohort 2) and a "validation" cohort (Cohort 3). We then randomly divided the discovery cohort samples into a training set (70%) and a testing set (30%), ensuring that disease status, HIV status, and cohort were equally represented (Fig. 1E). Next, we performed differential abundance analysis on the training set and filtered features using their adjusted $p$-value ($p < 0.05$) and base mean (mean > 100). We then trained 15 machine-learning classification models using the genes that were selected after filtering (Methods; Supplementary Table 1).

Most of the models performed well, with 8/15 models having a train and test area under the receiver operating characteristic curve (ROC-AUC) > 0.9 (Fig. 1F). However, some models, such as the generalized linear model (GLM) and linear discriminant analysis (LDA), suffered from poor test performance. The poor performance of the GLM may be due to the lack of regularization, since the performance of the generalized linear models with Ridge and LASSO feature selection (GLMNETRIDGE and GLMNETLASSO) were on par with the other models, suggesting that there may be several highly correlated features. LDA appeared to suffer from overfitting, possibly due to high collinearity in the features.

### 6-gene biomarker panel surpasses guidelines for a non-sputum-based triage test

After evaluating the performance of the different machine learning models, we chose the greedy forward search as it performed with the highest test set ROC-AUC. The greedy forward search algorithm is initiated by choosing the gene with the most discriminatory power in terms of a training score. This score is calculated as: (ROC-AUC + Sensitivity + Specificity) at the optimal Youden threshold. In each subsequent iteration, the algorithm adds the gene that produces the greatest increase in this score until there is no longer any gene which can increase the score by more than 0.01. This yielded a 6-gene signature consisting of guanylate binding protein 5 (*GBP5*), BCL2/adenovirus E1B interaction protein 3-like (*BNIP3L*), kruppel-like factor 6 (*KLF6*), Dysferlin (*DYSF*), LIM and SH3 protein 1 (*LASP1*), and poly(rC)-binding protein 1 (*PCBP1*) (Fig. 2A and Supplementary Table 2).

We evaluated the utility of the 6-gene signature as a diagnostic or triage tool using ROC analysis on the training, test, and validation sets. We found that the discriminatory power of the signature was high in all three sets using a singular cutoff (0.498) determined by the optimal Youden threshold on the training set (training set ROC-AUC = 0.921, test set ROC-AUC = 0.947, and validation set ROC-AUC = 0.946; Fig. 2B). In the training set, the TB score discriminated between TB positive and negative groups with 91.5% accuracy, 94.5% sensitivity, and 87.7% specificity. In the test set, the TB score discriminated between TB positive and negative groups with 91.8% accuracy, 97.1%

**Table 1 | Patient Information**

| | Cohort 1 | Cohort 2 | Cohort 3 |
|---|---|---|---|
| n | 93 | 98 | 60 |
| TB Positive, n (%) | 57 (61) | 50 (51) | 35 (58) |
| Age, median (IQR) | 29 (25-38) | 40 (28-54) | 31 (27-39) |
| Females, n (%) | 35 (38) | 41 (42) | 19 (32) |
| HIV Positive, n (%) | 16 (17) | 11 (11) | 10 (17) |
| Prior TB, n (%) | 13 (14) | 15 (15) | 7 (12) |

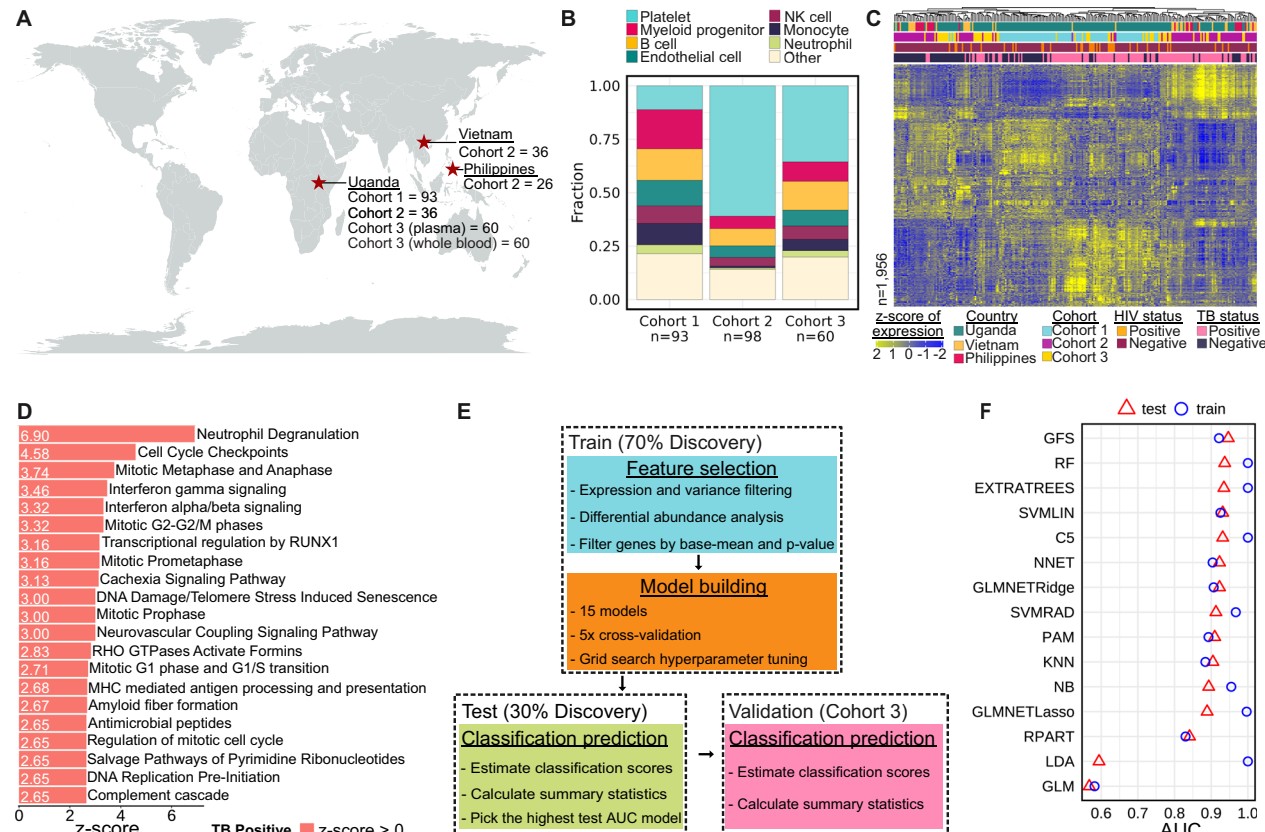

**Fig. 1 | Plasma cell-free RNA profiling. A** Geographic distribution of samples included in this study, which originated from three different cohorts. **B** Differences in cfRNA cell-types-of-origin between the three cohorts. Light-blue = Platelet, red = Myeloid-progenitor, yellow = B cell, dark-green = Endothelial cell, maroon = NK cell, dark-blue = Monocyte, light-green = Neutrophil, beige = Other. **C** VST normalized counts of significantly differentially abundant genes (two sided Wald test, Benjamini-Hochberg adjusted *p*-value < 0.05). Samples and genes are clustered based on correlation. Country: dark-green = Uganda, yellow = Vietnam, red = Philippines; Cohort: light-blue = Cohort 1, purple = Cohort 2, gold = Cohort 3; HIV status: orange = positive, maroon = negative; TB status: pink = positive, dark-blue = negative. **D** Top 21 differential pathways between microbiologically confirmed TB diagnoses ranked by significance. Z-score indicated in corresponding bar; direction relative to TB negative samples. **E** Flowchart of the method used to train, test, and validate the machine learning classification algorithms. **F** Area under the receiver operating characteristic curve (ROC-AUC) metrics for training and test sets across 15 machine learning models (test set = triangle, training set = circle).

sensitivity, and 85.2% specificity. Finally, in the validation set, the TB score discriminated between TB positive and negative groups with 88.3% accuracy, 97.1% sensitivity, and 76.0% specificity (Fig. 2C). Although the biomarker panel falls short of the specificity requirements for a diagnostic test, it exceeds the optimal criteria for a triage test on the test set. On the validation set, the panel exceeds the optimal sensitivity target for a triage test and falls 4 percentage points below the optimal specificity target, while still satisfying the minimal specificity target (optimal: 95% sensitivity, 80% specificity; minimal: 90% sensitivity, 70% specificity; Methods).

*GBP5* was identified as the gene with the strongest contribution to the performance of the 6-gene signature (Fig. 2A). We analyzed the effect of cohort and the association of *GBP5* with the Xpert MTB/RIF Ultra semi-quantitative result (Fig. 2D). We found that the expression level of *GBP5* in TB positive individuals is largely unaffected by cohort and sample collection (Supplementary Table 3). We also investigated the effect of country of origin and HIV status on the performance of the full 6-gene signature (Fig. 2E). We found that the removal of HIV-positive samples improved the performance of our model in both the test and validation set. This increase was largely due to the misclassification of HIV + /TB- patients, which accounted for 7 out of the 8 HIV+ samples that were misclassified. Notably, the signature retained a high accuracy for HIV + /TB+ individuals. Across a total of 18 HIV + /TB+ patients, the 6-gene signature misclassified only a single patient. To investigate the effect of country of origin, we calculated the ROC-AUC by country across all 251 samples and found that the model

performance was robust against the country of origin, with all three countries retaining an AUC greater than 0.9 (Uganda = 0.93, Vietnam = 0.90, Philippines = 0.96, Fig. 2F). Collectively these results demonstrate that the discriminatory power of the 6-gene signature is independent of geographic area and cohort and is moderately affected by HIV status.

Next, we investigated the correspondence of the 6-gene score with established markers of TB. The Xpert Ultra semi-quantitative result is a PCR-based test that bins samples based on the cycle threshold (CT) of the first positive probe that detects *M. tuberculosis* as follows: "low"=22< CT ≤ 38, "medium"=16< CT ≤ 22, and "high"=CT ≤ 16[14]. We found that the TB score was highly correlated with the Xpert Ultra semi-quantitative result in all three cohorts (polychoric correlation = 0.757; Fig. 2G and Supplementary Tables 4-5). Next, we compared the 6-gene signature to chest X-ray (CXR) scores which have been used as complementary information in determining the host response to TB. Recent developments in computer-aided detection (CAD) software have improved the performance of CXRs, but still fall short of the WHO TPPs for a triage test[15]. Three commercially available CAD applications were used to analyze CXRs from individuals enrolled in cohorts 2 and 3: CAD4TB (Delft Imaging, Netherlands), qXR (Quire.ai, India), and Lunit Insight chest X-ray TB algorithm (Lunit Inc., South Korea). The TB abnormality score is reported on a scale of 0-100 (for CAD4TB and Lunit) or 0-1 (for qXR), where a higher value indicates a more abnormal radiograph. Despite performance differences between each of the CAD algorithms, we found that the 6-gene TB score is

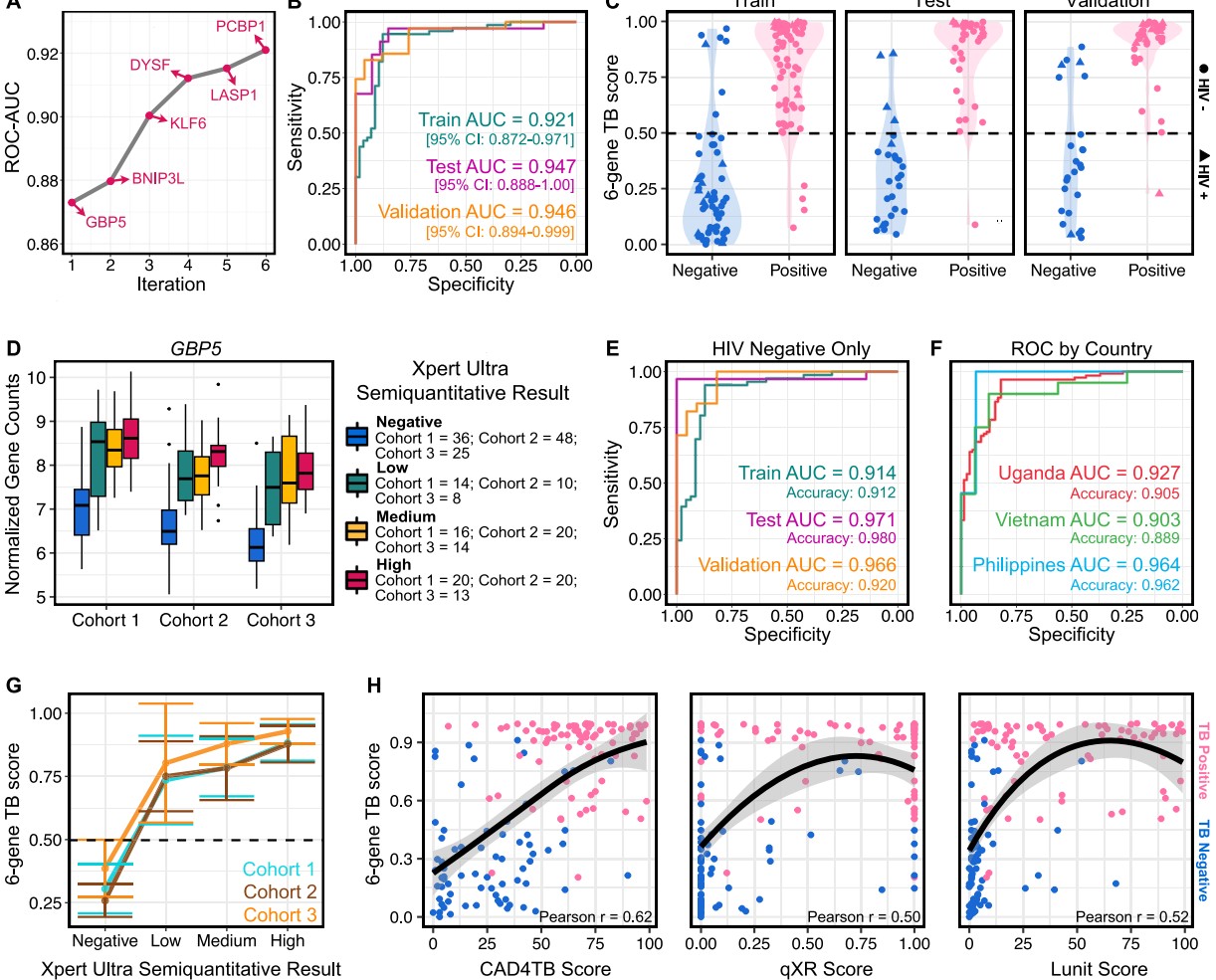

**Fig. 2 | Performance of the 6-gene signature identified in plasma cfRNA. A** Area under the receiver operating characteristic curve (ROC-AUC) as a function of the gene is added in each iteration of a greedy forward search model performed on the training dataset. **B** Train (dark-green), test (purple), and validation (gold) performance of the greedy forward search algorithm in distinguishing microbiologically confirmed TB. **C** Violin plot of classifier scores for training, test, and validation sets using the greedy forward search algorithm (TB positive = pink, TB negative = blue; HIV positive = triangle, HIV negative = circle). The dashed line represents the optimal Youden threshold cutoff (0.498) which was determined solely on the training set. **D** VST normalized counts of the most significant predictor *GBP5* across both cohort and semiquantitative TB status (blue = negative, dark-green = low,

yellow = medium, red = high). Boxes in the boxplots indicate the 25th and 75th percentiles, the band in the box represents the median, and whiskers extend to 1.5 x interquartile range of the hinge. **E** Performance of the 6-gene TB score when the model is re-evaluated without HIV-positive individuals. **F** Performance of the 6-gene TB score when separating samples by country (red = Uganda, green = Vietnam, light-blue = Philippines). **G** Correlation of the 6-gene TB score with the Xpert Ultra Semi-quantitative Result (dotted line: classification score threshold = 0.498; center: mean; bars indicate 95% confidence interval +/− SEM). Light-blue = Cohort 1, brown = Cohort 2, gold = Cohort 3. **H** Correlation of the 6-gene TB score with three chest X-ray scores. Color indicates disease status (pink = TB positive; blue = TB negative) Error bands represent the 95% CI around the LOESS smoothed line.

moderately correlated with the reported TB abnormality scores (Pearson r = 0.50-0.62; Fig. 2H), indicating that the gene expression signature may provide information associated with known metrics of host response. Collectively, these results indicate that the 6-gene signature has the potential to serve as a triage tool for TB.

### Comparison of plasma cfRNA biomarkers to whole blood RNA and protein studies

While many potential wbRNA signatures have been identified for the diagnosis of TB disease in clinically-relevant populations, only a handful meet the minimal WHO criteria for a triage test and none meet the minimal WHO criteria for a diagnostic test[4,16,17]. Unsurprisingly, the performance of these wbRNA signatures varies widely when applied to multi-country cohorts, as diagnostic performance heterogeneity between populations is a well-known barrier to the development of triage and diagnostic assays[18].

To compare the utility of cfRNA and wbRNA in relation to TB, we compared the 6-gene cfRNA signature to previously reported wbRNA

signatures. Among the wbRNA signatures that meet the minimal WHO criteria for a triage test, 7 have been evaluated for their performance in distinguishing TB from other diseases in a multi-cohort population (Berry86[19], daCosta2[20], daCosta3[20], Kaforou44[21], Walter47[22], Zak16[5], and Sweeney3[23]; Fig. 3A). We found that our 6-gene signature compares favorably to these wbRNA panels (Fig. 3B). We also noted that only one of the 6 genes in the cfRNA signature (*GBP5*) overlaps with those reported in previous wbRNA studies (Fig. 3A, Supplementary Data 4)[4].

To enable a direct comparison of the origin of cfRNA and wbRNA, we sequenced the wbRNA of 60 matched individuals from cohort 3 (mean reads: 13,521,651, range: 1,798,585 to 36,030,153). We then quantified the cell types that contribute RNA to the mixtures in plasma and whole blood samples collected from Cohort 3. We found that the origin of cfRNA and wbRNA is distinct (Fig. 3C). Plasma cfRNA is derived from both cells in the blood compartment and from vascularized solid tissues (solid organ fraction 0.23, 0.39, and 0.33 for Cohort 1, Cohort 2, and Cohort 3 respectively), while wbRNA is

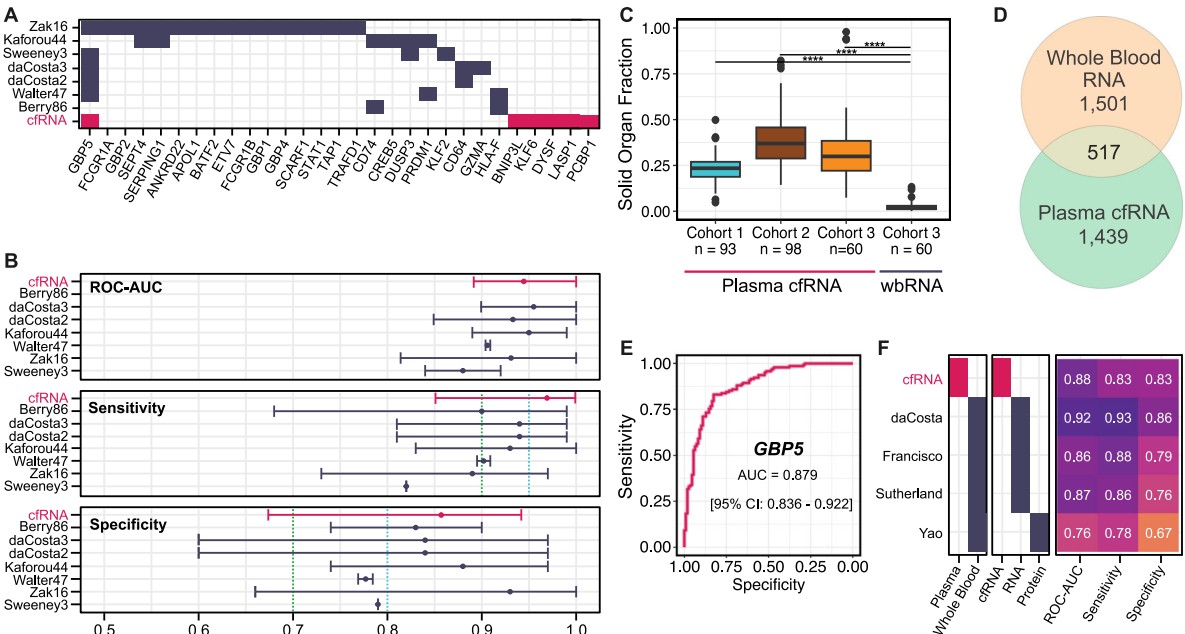

**Fig. 3 | Comparison of the 6-gene plasma signature with whole blood signatures of TB from literature. A** Overlap between top-performing whole blood signatures (dark-blue) and the 6-gene signature (pink). Top 29 overlapping genes are shown. **B** Performance comparison of whole blood signatures from previous multi-cohort studies (dark-blue) with our 6-gene cfRNA signature (pink). Optimal triage thresholds are marked in light-blue (95% sensitivity, 80% specificity), minimal triage thresholds are marked in green (90% sensitivity, 70% specificity). Error bars represent the 95% confidence intervals. Samples sizes (n) are as follows: cfRNA (61), Berry86 (116), daCosta2 (54), daCosta3 (54), Kaforou44 (103), Walter47 (180), Zak16 (49), Sweeney3 (787). **C** Comparison of solid organ fraction between the 3 cfRNA cohorts (light-blue, brown, gold) and the whole blood samples (dark-blue). "****" represents a two-sided Wilcoxin test, Benjamini-Hochberg adjusted $p$-value < 1e-15. Boxes in the boxplots indicate the 25th and 75th percentiles, the band in the box represents the median, and whiskers extend to $1.5 \times$ interquartile range of the hinge. **D** Venn diagram depicting the overlap of statistically significant, differentially expressed genes between whole blood and plasma cfRNA. **E** Performance of *GBP5* cfRNA abundance in distinguishing active TB. **F** Performance comparison of whole blood protein *GBP5*, whole blood RNA *GBP5*, and plasma cfRNA *GBP5* in distinguishing active TB.

predominately released by circulating immune cells (solid organ fraction of 0.025). To further evaluate the differences between cfRNA and whole blood RNA, we performed differential abundance analysis of cfRNA and wbRNA between TB positive and TB negative samples (DESeq2). We identified 2018 and 1956 differentially abundant genes in wbRNA and plasma cfRNA, respectively, 517 of which were shared (Benjamini-Hochberg adjusted $p$-value < 0.05, Fig. 3D; Supplementary Table 6; Supplementary Data 5). From our 6-gene signature, three genes were also identified as differentially abundant in our matched whole blood samples: *GBP5*, DYSF, and BNIP3L. This analysis provides evidence that plasma cfRNA can provide insights into mechanisms of cellular injury that are not accessible with wbRNA profiling.

*GBP5* is the gene most frequently reported across studies, occurring in our 6-gene cfRNA signature and in 5/7 whole blood signatures. *GBP5* has been shown to be critical for NLRP3 inflammasome activation, which mediates caspase-1 activation and secretion of proinflammatory cytokines in response to pathogenic bacteria and cellular damage[24,25]. Three wbRNA studies have found that *GBP5* alone has high discriminatory performance: da Costa et al.[20] (sensitivity = 93%, specificity = 86%, ROC-AUC = 0.924), Francisco et al.[26] (sensitivity = 88.2%, specificity = 78.5%, ROC-AUC = 0.86), and Sutherland et al.[27] which uses a Xpert-MTB-HR prototype with the Sweeney3[23] signature (sensitivity = 86%, specificity = 76%, ROC-AUC = 0.87). Similarly, we find that the abundance of *GBP5* in plasma cfRNA can separate TB and TB negative samples with a sensitivity of 83%, specificity of 83%, and an ROC-AUC of 0.88 (Fig. 3E and F). Whole blood *GBP5* protein levels have also been evaluated as a potential biomarker for TB, and have been demonstrated by Yao et al. to be significantly higher in people with microbiologically confirmed TB[28]. However, the performance of the blood protein biomarker at the optimal threshold (sensitivity = 0.78, specificity = 0.67, ROC-AUC = 0.76) is worse than

the plasma RNA biomarker. This suggests that plasma cfRNA may be a more effective biomarker for TB than whole blood *GBP5* protein levels.

## Discussion

In this study, we investigated the use of plasma cfRNA as a host-specific biomarker for the detection of TB. Using RNA sequencing and machine learning, we developed a 6-gene signature that can accurately distinguish individuals with microbiologically diagnosed TB. Our signature has a diagnostic accuracy, sensitivity, and specificity of 91.5%, 94.5%, and 87.7% in the training dataset, 91.8%, 97.1%, and 85.2% in the test dataset, and 88.3%, 97.1%, and 76.0% in the validation dataset. These metrics compare favorably to the performance of previous whole blood signatures. This is particularly important because non-sputum-based biomarker tests are a priority for TB management, as many high-risk populations are unable to produce high-quality sputum samples for diagnosis.

Our study is the first to investigate the potential of plasma cfRNA as a host-specific blood-borne biomarker for TB. Given that plasma cfRNA is a novel bioanalyte to monitor TB, these results are potentially important in at least three ways. First, our results suggest that signatures derived from plasma cfRNA may be more robust than those derived from whole blood RNA (wbRNA), which have shown poor performance in independent validation studies[4,17]. This may be due to the sensitivity of whole blood signatures to differences in patient cohorts and sample processing. We found that the plasma cfRNA signature is robust against these factors, making it a potentially more reliable source for TB biomarker discovery. Second, plasma cfRNA is stable, can be assayed using a small amount of plasma, and can provide insight into host cellular injury. Finally, future studies could explore the use of combined wbRNA and cfRNA bioanalytes for the development of more sensitive and specific diagnostic assays for TB.

Our study has several limitations that should be considered when interpreting the results. First, individuals were enrolled in this study under the presumption of having TB, and those diagnosed as not having TB reflect the local distribution of other conditions that also present with a cough of at least 2 weeks in duration. Cross-sectional studies including people with clinically diagnosed TB are needed to further validate accuracy. Second, the plasma sample collection method was not uniform across cohorts. While we retained high accuracy regardless of the collection tube and sample preparation method, further studies evaluating the impact of RNA preservatives and platelet contribution on the diagnostic performance may find improved performance. Third, we did not investigate patients with extrapulmonary TB, latent TB infection, or other forms of TB. Despite these limitations, our observations suggest that plasma cfRNA has the capability to serve as a proxy for tissue-specific changes in gene expression and may more adequately capture the host response to TB than wbRNA. Further research is needed to confirm these findings and develop a point-of-care, gene expression-based assay based on the plasma cfRNA signature. Such an assay could be a valuable tool in the early detection of TB and help improve the management and control of this disease.

## Methods

### Ethics statement

All patients provided written informed consent, and all experiments were performed in accordance with relevant guidelines and regulations. The protocols for this study were approved locally at each site by Institutional Review Boards at Cornell University (protocols IRB0145569, 1902008555); UCSF IRB 20-32670 (protocol 20-32670); University of Heidelberg Ethics Committee of the Medical Faculty (S-539/2020); the Makerere University, College of Health Sciences, School of Medicine, Research Ethics Committee 2020-182 (protocol 2017-020); Vietnam National Lung Hospital Ethical Committee for Biological Medical Research: 566/2020/NCKH (protocol 566/2020/NCKH), and De La Salle Health Sciences Institute Independent Ethics Committee 2020-33-02-A (protocol 2020-33-02-A).

### Sample collection

Plasma samples were collected as part of three different cohorts with separate sample collection methodologies. All three cohorts enrolled adults with cough ≥2 weeks identified at outpatient clinics in Uganda (Cohort 1 and Cohort 3) and in Uganda, Vietnam, and the Philippines (Cohort 2).

Peripheral blood samples from individuals enrolled in Cohort 1 were collected in Streck Cell-Free DNA blood collection tubes (Streck, 230257). Plasma was separated by centrifugation at $1600 \times g$ for 10 minutes at ambient temperature. The remaining cellular debris was removed by an additional centrifugation step at $16{,}000 \times g$ for 10 minutes at ambient temperature. Plasma was stored in 1 mL aliquots at −80 °C.

Plasma samples from individuals enrolled in Cohort 2 and Cohort 3 in sites in Uganda, Vietnam, and the Philippines were collected in K2EDTA blood collection tubes (BD Diagnostics, 366643). Plasma was separated by centrifugation at $1600 \times g$ for 10 minutes at ambient temperature in a horizontal rotor (swing out head). Plasma was similarly stored in 1 mL aliquots at −80 °C. Plasma sample volumes ranged from 100 μL to 1000 μL (Mean: 727 μL ± 227 μL, Supplementary Fig. 2). The plasma volume did not affect the measured cfRNA signature score (Supplementary Fig. 2).

Whole blood samples from individuals enrolled in the Cohort 3 in Uganda were collected in PAXgene Blood RNA tubes (Qiagen, 762165). The 2.5-mL PAXgene tubes were stored at −80 °C. Whole blood sample volumes were uniform at 700 μL.

### Plasma cfRNA isolation and library preparation

Plasma samples were received on dry ice and stored at −80 °C until processed. Prior to cfRNA extraction, plasma samples were thawed at room temperature and centrifuged at 1300 x $g$ for 10 minutes at 4 °C. cfRNA was extracted from plasma using the Norgen Plasma/Serum Circulating and Exosomal RNA Purification Mini Kit (Norgen, 51000). Extracted RNA was DNase treated with a combination of 10 μl DNase Turbo Buffer (Invitrogen, AM2238), 3 μl DNase Turbo (Invitrogen, AM2238), and 1 μl Baseline Zero DNase (Lucigen-Epicenter, DB0715K) for 30 minutes at 37 °C, then concentrated into 12 μl using the Zymo RNA Clean and Concentrated Kit (Zymo, R1015).

Sequencing libraries were prepared from 8 μl of concentrated RNA using the Takara SMARTer Stranded Total RNA-Seq Kit v3 - Pico Input Mammalian (Takara, 634485) and barcoded using the SMARTer RNA Unique Dual Index Kit (Takara, 634451). Library concentration was quantified using the Qubit 3.0 Fluorometer (Invitrogen, Q33216) with the dsDNA HS Assay Kit (Invitrogen, Q32854). Libraries were quality-controlled using the Agilent Fragment Analyzer 5200 (Agilent, M5310AA) with the HS NGS Fragment Kit (Agilent, DNF-474-0500) and pooled to equal concentrations. Each pool was sequenced using both the Illumina NextSeq 500/550 platform (paired-end, 150 bp) and the Illumina NextSeq 2000 platform (paired-end, 100 bp).

### Whole blood RNA isolation and library preparation

Whole blood samples were received on dry ice and stored at −80 °C until processed. Prior to RNA extraction, whole blood samples were thawed at 37 °C. RNA was extracted from whole blood samples using the Quick-RNA Whole Blood Kit (Zymo, R1201) following the manufacturer's instructions. RNA was processed using the NEBNext Globin & rRNA Depletion Kit (Human/Mouse/Rat; NEB, E7750X), and libraries were created using the NEBNext Ultra II Directional RNA Library Prep Kit (NEB, E7760 and E7765) following manufacturer's specifications.

Sequencing libraries were quantified using the Qubit 3.0 Fluorometer (Invitrogen, Q33216) with the dsDNA HS Assay Kit (Invitrogen, Q32854). Libraries were quality-controlled using the Agilent Fragment Analyzer 5200 (Agilent, M5310AA) with the HS NGS Fragment Kit (Agilent, DNF-474-0500) and pooled to equal concentrations. Each pool was sequenced using the Illumina NextSeq 2000 platform (paired-end, 100 bp).

### Bioinformatic processing and sample quality filtering

Sequencing data was processed using a custom bioinformatics pipeline. Since pools sequenced on the Illumina NextSeq 2000 were optimized to produce paired-end, 61 bp reads, matched sequencing data from the Illumina NextSeq 500 were trimmed using Seqtk (v1.2). Samples were then quality filtered and trimmed using BBDuk (v38.90), and aligned to the Gencode GRCh38 human reference genome (v38, primary assembly) using STAR[29] (v2.7.0 f, default parameters). Prior to feature quantification using featureCounts[30] (v2.0.0), samples were deduplicated using Picard MarkDuplicates (v2.19.2). Mitochondrial, ribosomal, X and Y chromosome genes were bioinformatically removed prior to analysis.

Samples were filtered on the basis of DNA contamination, rRNA contamination, total counts, and RNA degradation. DNA contamination was estimated by calculating the ratio of reads mapping to introns and exons. rRNA contamination was measured using SAMtools (v1.14). Total counts were calculated using featureCounts[30] (v2.0.0). Degradation was estimated by calculating the 5′-3′ bias using Qualimap[31] (v2.2.1). Samples were removed from analysis if either the intron to exon ratio was greater than 4, if a sample had less than 90,000 total feature counts, or if the rRNA contamination, total counts, or 5′-3′ bias was greater than 4 standard deviations from the mean.

### Cell type deconvolution

Cell-type deconvolution was performed using BayesPrism[9] (v1.1) with the Tabula Sapiens single-cell RNA-seq atlas[10] (Release 1) as a reference. Cells from the Tabula Sapiens atlas were grouped as previously described in Voreperian et al.[8]. Cell types with more than 100,000

unique molecular identifiers (UMIs) were included in the reference and subsampled to 300 cells using ScanPy[32] (v1.8.1).

### Differential abundance analysis

Comparative analysis of differentially expressed genes was performed using a negative binomial model as implemented in DESeq2[11] (v1.34.0). Heatmaps were constructed using the pheatmap package in R (v1.0.12). Samples and genes were clustered using correlation-based hierarchical clustering. Canonical pathways, diseases and functions were analyzed using QIAGEN Ingenuity Pathway Analysis software (v73620684).

### Machine learning and model training

Machine learning and model training were performed using R (v4.1.3) with the DESeq2[11] (v1.34.0), Caret[33] (v6.0.90), and pROC[34] (v1.18.0) packages. Sample metadata and count matrices were split 70/30 into a training set and a test set. To ensure a representative split, we assigned a label to each sample indicating its status for the variables of cohort, HIV status, and TB status. We then distributed samples into train and test using this label to ensure equal representation of cohort, HIV, and TB in both sets. The validation set ($n = 60$) was set aside for the evaluation of our model performance.

Features for model training were selected using differential abundance analysis. We excluded genes with a base mean of less than 100 and a Benjamini-Hochberg adjusted $p$-value greater than 0.05. The remaining 113 genes were selected for model training. Machine learning algorithms were trained using 5-fold cross-validation and grid search hyperparameter tuning. Accuracy, sensitivity, specificity, and area under the receiver operating characteristic curve (ROC-AUC) were used to measure test performance. The classification models used were generalized linear models with Ridge and LASSO feature selection (GLMNETRIDGE and GLMNETLASSO), support vector machines with linear and radial basis function kernel (SVMLin and SVMRAD), random forest (RF), random forest ExtraTrees (EXTRATREES), neural networks (NNET), linear discriminant analysis (LDA), nearest shrunken centroids (PAM), C5.0 (C5), k-nearest neighbors (KNN), naive bayes (NB), CART (RPART), and generalized linear models (GLM). An additional model was trained and tested using a greedy forward search algorithm (GFS). Briefly, this algorithm iterates over the list of genes and evaluates each individual gene's discriminatory power by training a generalized linear model (GLM). The discriminatory power is evaluated based on a combined score, which is calculated as: (AUC + Sensitivity + Specificity), in which sensitivity and specificity are calculated at the optimal yield threshold. The gene that results in the highest score is added to the model. At each subsequent iteration over the list of candidate genes, the algorithm will attempt to add a gene to the model in this way. The algorithm stops when there is no gene in the list that can increase the score by more than 0.01.

### WHO TPP thresholds

The broader diagnostic potential of our 6-gene signature was informed by the WHO target product profiles (TPP's). The WHO TPP's provide the minimal and optimal characteristics of a target product, where the minimal thresholds indicate the lowest acceptable standard, and the optimal criteria outline the ideal target profile. For a non-sputum-based diagnostic test, the overall minimal and optimal sensitivities are 65% and 80% with a specificity of 98%. For a non-sputum-based triage test, the optimal requirements are 95% sensitivity and 80% specificity, while the minimal criteria are 90% sensitivity and 70% specificity.

### Quantification and statistical analyzes

All statistical analyses were performed using R (v4.1.0). Statistical significance was tested using Wilcoxon signed-rank tests, and Mann-Whitney U tests in a two-sided manner unless otherwise stated. Boxes in the boxplots indicate the 25th and 75th percentiles, the band in the

box represents the median, and whiskers extend to 1.5 × interquartile range of the hinge. All sequencing data was aligned to the GRCh38 Gencode v38 Primary Assembly, and features were counted using the GRCH38 Gencode v38 Primary Assembly Annotation.

### Reporting summary

Further information on research design is available in the Nature Portfolio Reporting Summary linked to this article.

## Data availability

The raw sequencing data and de-identified RNA-seq count matrices generated in this study have been deposited in the Gene Expression Omnibus under the accession codes GSE255071, GSE255073, GSE255074. All data are included in the Supplementary Information or are available from the authors, as are unique reagents used in this Article. The raw numbers for charts and graphs are available in the Source Data file whenever possible. Source data are provided in this paper.

## Code availability

All code is available on GitHub at (https://github.com/DanielEweisLaBolle/cfRNA_TB).

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

## Acknowledgements
We thank the Cornell Genomics Center and the UCSF Center for Advanced Technology for help with sequencing libraries. We thank the patients and their families for contributing their blood to further our understanding of TB. This work was supported by the National Institutes of Health (NIH) grants R01AI146165, R21AI133331, R21AI124237, R01AI151059. The funders had no role in study design, data collection and analysis, decision to publish, or preparation of the manuscript.

## Author contributions
A. Chang, A.S., and I.D.V. contributed to the study design. J.S.L. performed the experiments. A. Chang, C.J.L, D.E.L, and I.D.V. analyzed the data. A.S., A. Cattamanchi, A.A., N.V.N., C.Y., W.W., C.M.D., and P.N. facilitated data collection. A. Chang, C.J.L, D.E.L, and I.D.V. wrote the manuscript. All authors provided comments and edits.

## Competing interests
A. Chang, C.J.L, D.E.L, and I.D.V are inventors on submitted patents pertaining to cell-free nucleic acids (US patent applications 63/237,367 and 63/429,733). I.D.V. is a member of the Scientific Advisory Board of Karius Inc., Kanvas Biosciences, and GenDX. I.D.V. is listed as an inventor on submitted patents pertaining to cell-free nucleic acids (US patent applications 63/237,367, 63/056,249, 63/015,095, 16/500,929, 41614P-10551-01-US) and receives consulting fees from Eurofins Viracor. The remaining authors declare no competing interests.
