## [Peer Review File · Nature Communications]

REVIEWER COMMENTS

Reviewer #1 (see attached document)

Reviewer #2 (Remarks to the Author):

In general, this is a novel analysis which needs to be validated, but has the potential to improve TB diagnosis. Previous blood based RNA signatures were unable to be validated at high sensitivity in external validation cohorts. (Maybe because of the heterogeneity of the disease as Canetti explained in 1955 or maybe due to the non discrete spectrum of the TB syndrome, or overlapping immune response with other diseases). But this is exciting preliminary work. Why will cfRNA not fall into the same problems that beleaguered cell based transcriptomics?

In summary, they evaluated 182 individuals w cough > 2 weeks and found 100 w micro confirmed TB (Uganda, Vietnam, Philippines). Then, they used plasma and evaluated cell-free transcriptomics (cfRNA) and identified 541 differentially abundant genes between those with TB and those without TB. (What are the non-TB diseases the "other disease" group? This should be listed with clarify in the supplemental tables. Could some of them still have had TB? Were any give TB medications despite not having a microbiologic diagnosis?)

The limited machine learning techniques to top 150 used to train 15 machine learning models

The highest performing model was a 9 gene signature: diagnostic accuracy of 89% and AUC 0.93

A minor criticism: there is no external validation dataset (because they are the first to do such a study on cfRNA); this is a minor limitation and should be mentioned that this needs to be validated in external cohorts. (the 2 cohorts were divided 70/30 into train and test sections. (Are 30 TB cases enough for accurate testing?))

Line 81/ pg 2: A contaminated MIGIT was considered micro confirmed? Why?

It would be nice to see a comparison of DEG in cfDNA versus what has been reported in previous papers (Berry, Gupta, Warsinske, etc). This is nicely depicted in Fig 3a; thank you; Is a full comparison included in the supplemental table 7?

Figure 1D legend: please mention what pathway analysis was used. (IPA, yes?) Thank you for providing the z-score- very helpful to give directionality. But the figure legend doesn't give the precise directionality. I am assuming z-score <0 is Tb over controls, meaning EIF2 is decreased in TB pts compared to controls, while pyroptosis and ephrin are increase in TB over controls? Please make this more explicit.

Again, I think 1D is extremely helpful and very well represented. The authors are applauded for not reporting only enrichment, but also giving z-score directionality. That being said, very few labs can afford

IPA and therefore the results are not reproducible to those who can't afford IPA. Please provide a supplemental table of the KEGG, GO, and Hallmark pathways for researchers who can't afford the proprietary IPA. When using IPA pathways, a supplemental table should be provided with the enriched genes and the total genes in each pathway. (Otherwise, how can the results be validated? Except by those with super funding to afford IPA?)

Figure 1B is helpful. Why do you think polymorphic neutrophils are not represented more considering they are >50% of circulating immune cells? I agree that cfRNA will better represent organ specific pathology and dying cells. By where are the neutrophils?

Figure 1F is also very well done; since culture and Xpert are far from perfect, do you think the test needs to have perfect specificity? How did the different signatures compare against a clinical diagnosis? (The methods do not state if culture and Xpert were performed on a single sputum or 3 sputums? Can you clarify that? See comment above about misclassification and clinical diagnosis)

Figure 3 only has an A, B, C, D and E, yet there are 6 sub-figures. Please label the "pyroptosis and necrosis-specific markers" sub-figure and describe it in the figure legend. Why is MARCO not included in 3a?

For Figure 3B, in the figure legend, can you please clarify where these numbers are coming from? From the original publications or from the authors re-analysis?

Figure 3C is missing a legend. What is pink and what is dark blue?

The data availability is in the future tense: the Github code and raw data should be entered into GEO at the time of peer review so that reviewers can replicate some of the analysis and confirm some preliminary findings. A decent bioinformatician should be able to replicate the main results easily; this should be feasible before review of the revisions of the very nice paper.

Reviewer #3 (Remarks to the Author):

This article is about the use of cfRNA in plasma as source of diagnostic biomarkers for a non-sputum based test for the detection of tuberculosis. Through an RNA profiling by NGS followed by a machine learning approach, authors identify a 9-gene signature whose diagnostic performance outperform that of some whole-blood RNA based signatures and fulfil the criteria for a non-sputum triage test. The quality of the manuscript is provided by controlled experimental procedures and by a clear exposure of the results. Some minor changes could improve the overall quality of the manuscript.

Main concerns:

-in the method section it is written that samples of the END TB cohort have been collected using Streck cell-free DNA collection tubes, I wonder why specific tubes for RNA preservation have not been used instead of those for DNA.

-in methods section, cfRNA isolation paragraph, a range of volumes of plasma has been reported for the RNA isolation, does this mean that RNA has been isolated from different volumes of plasma in different

samples? If so authors should explain why. Could this differences have an impact on the quantification RNAs detected?

-At page 7, it is not clear how the Authors suggest to complement the chest X-ray results with the analysis of the 9-gene signature. Authors could explain better this aspect in the discussion section.

Minor concerns:

-the specification of some acronyms, such as those of the algorithms used, could improve the reading of the results

- at page 5, line 157, minimal and optimal sensitivities may have been inverted.

-at page 6, a reference about the thresholds used in the Xpert method is appropriate.

Comments on the manuscript entitled, “*Circulating cell-free RNA in blood as a host response biomarker for detection of Tuberculosis*” Chang et al.

As there are no sensitive tests available to diagnose different stages of TB infection, like subclinical, progressive and latent infection, the authors have carried out a study using circulating cell-free (CfRNA) (mostly released by dying cells) as host-based transcriptomic signatures as biomarkers for the diagnosis of tuberculosis.

The authors selected 182 individuals who were having a duration of cough of ≥ 2 weeks, from 2 separate clinical trials, named ENDTB and R2D2, conducted in Uganda, Vietnam and the Philippines. Plasma samples were collected from these patients for the study. Of these participants, 100 individuals were diagnosed microbiologically as confirmed TB.

CfRNA and library preparations were performed by standard methods. Bioinformatic processing and sample quality filtering were performed by employing standard techniques. Differential abundance analysis was done by using recommended software. Machine learning and model training was performed using recommended software. Statistical analysis was performed using standard methods.

The study showed 9 gene signatures (GBPS, DYSF, SMARCD3, WAMP5, FCGR3B, GPI, MPO, CREB5, GBP2) having a diagnostic efficacy of 89.1% with sensitivity of 0.962 and specificity of 0.897 and could have the potential in developing as a blood-based triage test for diagnosis of TB.

Figures and tables have been properly placed with adequate labelling.

Work has been carried out meticulously using standard methodology and techniques. A piece of very new information has been obtained based on the host's response to the infection. The study is the first report on a cfRNA-based technology using plasma samples from patients.

Although the data presented here seems interesting, a few questions arise and need to be clarified:-

The study should have been further applied in a cohort-designed study to find out the variations in response. Further, it should be done in all forms of TB, especially in extrapulmonary TB and latent infection.

The references are not comparable with the text after reference number 14. It is suggested to correct the sequence and number according to the text. Reference No. 15 (WHO reference) has not been added to the text.

Whole blood RNA is not only shed from living cells but to some extent from dying cells as well. Hence, is it true to say that cell death pathways and mechanism of cellular injury cannot be obtained from wbRNA profiling. It needs justification.

How samples have been divided into a training set (70%) and a test set (30%)?

What is the possible reason that the GPI gene normalized count is not robust against differences in sample collection on evaluating it individually?

In terms of computer-aided detection of CXR of TB. These are not highly specific then how this 9-gene signature model can provide complementary information in the evaluation of TB?

It is an extensive study performed to reach a conclusion that the 9 gene model can be highly accurate and specific for TB diagnosis in non-sputum-based assays. One major limitation of the study is that they have not performed and shown differential gene expression in whole blood. This analysis can further validate the 9-gene model or else some more specific genes can be identified and included in TB diagnosis.

English language is lucid and it was understandable to the reader.

The manuscript should be accepted for publication after major revision.

Point-by-point address of the specific comments raised by the editors and reviewers.

Reviewer #1 (Remarks to the Author):

Comments on the manuscript entitled, "Circulating cell-free RNA in blood as a host response biomarker for detection of Tuberculosis" Chang et al.

As there are no sensitive tests available to diagnose different stages of TB infection, like subclinical, progressive and latent infection, the authors have carried out a study using circulating cell-free (CfRNA) (mostly released by dying cells) as host-based transcriptomic signatures as biomarkers for the diagnosis of tuberculosis.

The authors selected 182 individuals who were having a duration of cough of ≥ 2 weeks, from 2 separate clinical trials, named ENDTB and R2D2, conducted in Uganda, Vietnam and the Philippines. Plasma samples were collected from these patients for the study. Of these participants, 100 individuals were diagnosed microbiologically as confirmed TB.

CfRNA and library preparations were performed by standard methods. Bioinformatic processing and sample quality filtering were performed by employing standard techniques. Differential abundance analysis was done by using recommended software. Machine learning and model training was performed using recommended software. Statistical analysis was performed using standard methods.

The study showed 9 gene signatures (GBPS, DYSF, SMARCD3, WAMP5, FCGR3B, GPI, MPO, CREB5, GBP2) having a diagnostic efficacy of 89.1% with sensitivity of 0.962 and specificity of 0.897 and could have the potential in developing as a blood-based triage test for diagnosis of TB.

Figures and tables have been properly placed with adequate labeling.

Work has been carried out meticulously using standard methodology and techniques. A piece of very new information has been obtained based on the host's response to the infection. The study is the first report on a cfRNA-based technology using plasma samples from patients.

Thank you for the careful analysis of the paper and thank you for your appreciation of this work. In the revised manuscript, we have addressed all your comments and suggestions, and we include data from extensive new experiments and analyses. Most importantly, we include analysis of 69 additional cfRNA samples, including 60 new cfRNA samples from a validation cohort that we used as an unbiased way to evaluate the performance of the model. We developed a new model, which yielded a new 6-gene signature for tuberculosis which achieved a high performance in differentiating TB positive and negative individuals: with an Area Under the Curve of 0.92, 0.95 and 0.95 in the training, testing and validating sets, respectively (compared to a measured AUC of 0.93 in the test set in the original manuscript). The change in marker genes from the originally reported panel is due to increasing the sample set and a new TB scoring method for the GFS algorithm which considers AUC, sensitivity, and specificity while training the model. Last, we

include new data and analyses of 60 whole blood RNA samples from the same cohort. This new experiment enabled us to investigate the differences in origin of wbRNA and plasma cfRNA. We believe these revisions and new data significantly improve our manuscript and expand the merit of our findings. We note that for clarity, we have renamed the cohorts END TB, R2D2 and TB2 to Cohort 1, Cohort 2 and Cohort 3 respectively.

Although the data presented here seems interesting, a few questions arise and need to be clarified:

1. The study should have been further applied in a cohort-designed study to find out the variations in response. Further, it should be done in all forms of TB, especially in extrapulmonary TB and latent infection.

With this case-control study we are the first to report on cell-free RNA in blood plasma as a potential novel class of host response biomarker for TB. In the revised manuscript, we include data for an independent validation cohort, expanding the scope of our study further to include samples from three clinical studies, three countries and 251 individuals (up from 182 individuals in the original manuscript). In addition, we include a novel analysis of plasma and whole blood from the same cohort to further the understanding of the utility of cfRNA-based transcriptomics for TB diagnostics. We believe this is a very large scope for a proof-of-concept study. We agree with the reviewer that future studies should test these novel principles and to investigate cfRNA biomarkers of extrapulmonary, latent, and incipient TB. As part of this study, we were unable to obtain samples representing all forms of TB, given that samples were not collected with this goal in mind. We address this limitation in the discussion section.

The references are not comparable with the text after reference number 14. It is suggested to correct the sequence and number according to the text. Reference No. 15 (WHO reference) has not been added to the text.

Thank you for pointing this out. We have corrected this in the revised manuscript.

2. Whole blood RNA is not only shed from living cells but to some extent from dying cells as well. Hence, is it true to say that cell death pathways and mechanism of cellular injury cannot be obtained from wbRNA profiling. It needs justification.

The reviewer raises an excellent point. To address this, we have performed additional experiments on plasma and whole blood samples, which allowed us to measure the differences in origin of cfRNA and wbRNA. We find that the proportion of cfRNA derived from solid tissues far exceeds the proportion of wbRNA derived from solid tissues, supporting the principle that cfRNA and wbRNA have a different origin. The point of the reviewer is however well taken, it is expected that to an extent wbRNA is also derived from dying cells, and we have therefore rephrased this in the revised manuscript.

3. How samples have been divided into a training set (70%) and a test set (30%)?

Samples from the discovery set (n=191) were randomly split 70/30 in training (n = 130) and test (n = 61) sets. We assigned a label to each sample indicating its status for the variables of cohort, HIV status and TB status. We then distributed samples into train and test using this label so as to ensure equal representation of cohort, HIV and TB in both sets. The validation set (n=60) was set aside for evaluation of our model performance. We have clarified this in the Methods section.

4. What is the possible reason that the GPI gene normalized count is not robust against differences in sample collection on evaluating it individually?

Differences in sample collection significantly affect the cell types of origin of cfRNA (see Figure 1B), which some genes are more sensitive to. Future studies should implement uniform sample collection and plasma isolation protocols, which we expect will lead to even better performance of cfRNA as host response biomarker of TB.

5. In terms of computer-aided detection of CXR of TB. These are not highly specific then how this 9-gene signature model can provide complementary information in the evaluation of TB?

We agree with the reviewer that computer-aided detection CXR for TB is not a highly-specific method of diagnosis. Whether plasma cfRNA profiling provides complementary information should be evaluated in future studies (ours was not designed to compare the accuracy of the two methods). For example, future studies may assess correlation between genes or gene signatures and CXR measurements (e.g. cavitation, opacity, texture, and shape). However, a major challenge is that CXR equipment is not routinely available in high burden countries. Our hope is that a plasma cfRNA signature can be more easily translated into a widely available, point-of-care test.

6. It is an extensive study performed to reach a conclusion that the 9 gene model can be highly accurate and specific for TB diagnosis in non-sputum-based assays. One major limitation of the study is that they have not performed and shown differential gene expression in whole blood. This analysis can further validate the 9-gene model or else some more specific genes can be identified and included in TB diagnosis.

To address this comment, we have updated our analysis to include differential gene expression of tuberculosis in both whole blood RNA and plasma cfRNA. We show the overlap in TB related differentially abundant genes between both biofluid types. We also show that 3 of the 6 genes in our new biomarker panel are identified as differentially abundant in plasma cfRNA.

English language is lucid and it was understandable to the reader. The manuscript should be accepted for publication after major revision.

Reviewer #2 (Remarks to the Author):

In general, this is a novel analysis which needs to be validated, but has the potential to improve TB diagnosis. Previous blood based RNA signatures were unable to be validated at high sensitivity in external validation cohorts. (Maybe because of the heterogeneity of the disease as Canetti explained in 1955 or maybe due to the non discrete spectrum of the TB syndrome, or overlapping immune response with other diseases). But this is exciting preliminary work. Why will cfRNA not fall into the same problems that beleaguered cell based transcriptomics?

In summary, they evaluated 182 individuals w cough > 2 weeks and found 100 w micro confirmed TB (Uganda, Vietnam, Philippines). Then, they used plasma and evaluated cell-free transcriptomics (cfRNA) and identified 541 differentially abundant genes between those with TB and those without TB.

Thank you for your appreciation of our work. Your comments and suggestions have allowed us to further improve our analysis and the presentation of our results. We note that, in the revised manuscript, we include data from extensive new experiments and analyses. Most importantly, we include analysis of 69 additional cfRNA samples, including 60 new cfRNA samples from a validation cohort that we used as an unbiased way to evaluate the performance of the model. Based on these data, we developed a 6-gene signature for tuberculosis which achieved a high performance in differentiating TB positive and negative individuals: with an Area Under the Curve of 0.92, 0.95 and 0.95 in the training, testing and validating sets, respectively (compared to a measured AUC of 0.93 in the test set in the original manuscript). The change in marker genes from the originally reported panel is due to increasing the sample set and a new TB scoring method for the GFS algorithm which considers AUC, sensitivity, and specificity while training the model. Last, we include new data and analyses of 60 whole blood RNA samples from the same cohort. This new experiment enabled us to investigate the differences in origin of wbRNA and plasma cfRNA. We believe these revisions and new data significantly improve our manuscript and expand the merit of our findings. We note that for clarity, we have renamed the cohorts END TB, R2D2 and TB2 to Cohort 1, Cohort 2 and Cohort 3 respectively.

1. What are the non-TB diseases in the "other disease" group? This should be listed with clarity in the supplemental tables. Could some of them still have had TB? Were any given TB medications despite not having a microbiologic diagnosis?

Follow-up diagnoses were not performed in the "other disease" groups. Samples were drawn from a representative population of people with presumptive TB. Individuals in the "other disease" groups reflect the local distribution of other conditions that also present with a cough of >2 weeks duration. 9 people in the TB-negative group were treated for TB. Empiric treatment for TB is common in high burden settings (and treatment decisions were made before culture results were available). We have added to the limitations paragraph of the Discussion that future studies should include people clinically diagnosed with TB

(ideally after careful follow-up to assess response to empiric TB treatment or symptom resolution without TB treatment).

2. The limited machine learning techniques to top 150 used to train 15 machine learning models. The highest performing model was a 9 gene signature: diagnostic accuracy of 89% and AUC 0.93. A minor criticism: there is no external validation dataset (because they are the first to do such a study on cfRNA); this is a minor limitation and should be mentioned that this needs to be validated in external cohorts. (The 2 cohorts were divided 70/30 into train and test sections. (Are 30 TB cases enough for accurate testing?))

To address this comment we include data for an independent validation cohort of 60 plasma cfRNA samples from patients enrolled at outpatient clinics in Uganda in the new version of the manuscript. We agree with the reviewer that more extensive validation is needed, however, the high performance of our 6-gene signature on this validation cohort (AUC 0.95) is promising in this regard. We note that the 70/30 split refers to the percentage of samples included in the train and test sets.

3. Line 81/ pg 2: A contaminated MGIT was considered micro confirmed? Why?

The parent study SOPs included performing Xpert testing on sediment from MGIT culture tubes that were reported as contaminated (*i.e.*, had bacterial growth that stained negative with Ziehl-Nielsen staining). This technique has been shown to improve detection of TB within MGIT tubes reported as contaminated: [https://www.thelancet.com/journals/lanmic/article/PIIS2666-5247\(23\)00169-6/fulltext](https://www.thelancet.com/journals/lanmic/article/PIIS2666-5247(23)00169-6/fulltext)

However, none of the patients included in this analysis were diagnosed with TB based on Xpert on contaminated MGIT (or on urine Xpert results). Therefore, we have simplified the methods to indicate that all TB diagnoses were based on sputum Xpert Ultra and culture results.

4. It would be nice to see a comparison of DEG in cfDNA versus what has been reported in previous papers (Berry, Gupta, Warsinske, etc). This is nicely depicted in Fig 3a; thank you; Is a full comparison included in the supplemental table 7?

Supplementary Table 7 shows the comparison of lung-, pyroptosis-, and necrosis-specific gene abundances (genes identified in Figure 3C) in TB positive versus TB negative groups for each cohort. We have included the full comparison of gene signatures identified in cfDNA versus previous whole blood papers in Supplementary Data File 4.

5. Figure 1D legend: please mention what pathway analysis was used. (IPA, yes?) Thank you for providing the z-score- very helpful to give directionality. But the figure legend doesn't give the precise directionality. I am assuming z-score <0 is Tb over controls, meaning EIF2 is decreased in TB pts compared to controls, while pyroptosis and ephrin are increased in TB over controls? Please make this more explicit.

We updated the figure caption to make it clearer that z-score directionality is TB positive versus TB negative.

6. Again, I think 1D is extremely helpful and very well represented. The authors are applauded for not reporting only enrichment, but also giving z-score directionality. That being said, very few labs can afford IPA and therefore the results are not reproducible to those who can't afford IPA. Please provide a supplemental table of the KEGG, GO, and Hallmark pathways for researchers who can't afford the proprietary IPA. When using IPA pathways, a supplemental table should be provided with the enriched genes and the total genes in each pathway. (Otherwise, how can the results be validated? Except by those with super funding to afford IPA?)

We have added a Supplementary Data File 2, which contains the IPA pathway output. We have also performed gene set enrichment analysis using KEGG, GO, and Hallmark pathways as suggested. The results are concordant with the IPA results and are included in Supplementary Data File 3.

7. Figure 1B is helpful. Why do you think polymorphic neutrophils are not represented more considering they are >50% of circulating immune cells? I agree that cfRNA will better represent organ specific pathology and dying cells. By where are the neutrophils?

The burden of cfRNA of a specific cell type depends on the quantity of these cells in the circulation and their turnover rate. Neutrophils are highly abundant, but have a relatively low turnover rate.

8. Figure 1F is also very well done; since culture and Xpert are far from perfect, do you think the test needs to have perfect specificity? How did the different signatures compare against a clinical diagnosis? (The methods do not state if culture and Xpert were performed on a single sputum or 3 sputums? Can you clarify that? See comment above about misclassification and clinical diagnosis)

For this case-control study, all cases (ie, TB group) had positive sputum Xpert or culture result. Both tests are known to be highly specific (>98.5% specificity), meaning that a positive result indicates TB. Both are considered confirmatory tests for TB and recommended as such by the World Health Organization.

All controls (ie, "other disease" group) had one negative Xpert and two negative culture results at baseline. The tests were performed on either 2 or 3 sputum specimens depending on the site. We pre-defined a microbiological reference standard and therefore classified TB status based on sputum microbiologic tests alone.

We agree that a plasma cfRNA test does not have to have perfect specificity. The most likely use case is as a TB screening or triage test, for which the WHO recommends

minimum sensitivity of 90% and minimum specificity of 70%. We show that plasma cfRNA signatures exceed these thresholds. We have added to the limitations section of the Discussion that these results need further validation among patients clinically diagnosed with TB (despite negative sputum Xpert and culture results).

9. Figure 3 only has an A, B, C, D and E, yet there are 6 sub-figures. Please label the “pyroptosis and necrosis-specific markers” sub-figure and describe it in the figure legend. Why is MARCO not included in 3a?

We have re-formatted figure 3 and have moved this analysis to supplementary figure 4. We made this change because the new data and analysis of wbRNA and cfRNA profiles allows us to more directly demonstrate the differences in origin of these different classes of blood derived RNA (Figures 3C and 3D)

10. For Figure 3B, in the figure legend, can you please clarify where these numbers are coming from? From the original publications or from the authors re-analysis?

The numbers from the whole blood signatures were obtained from the publications. We have clarified this in the main text and in the figure legend.

11. Figure 3C is missing a legend. What is pink and what is dark blue?

We have bolded the legend in Figure 3C to make it more obvious, and included the legend in the figure caption. In Figure 3C, pink indicates TB positive samples and dark blue indicates TB negative samples.

12. The data availability is in the future tense: the Github code and raw data should be entered into GEO at the time of peer review so that reviewers can replicate some of the analysis and confirm some preliminary findings. A decent bioinformatician should be able to replicate the main results easily; this should be feasible before review of the revisions of the very nice paper.

The GitHub code and count matrices were uploaded during the review process, and can be accessed at https://github.com/DanielEweisLaBolle/cfRNA_TB. We have updated the text in the methods section.

Reviewer #3 (Remarks to the Author):

This article is about the use of cfRNA in plasma as a source of diagnostic biomarkers for a non-sputum based test for the detection of tuberculosis. Through an RNA profiling by NGS followed by a machine learning approach, authors identify a 9-gene signature whose diagnostic performance outperform that of some whole-blood RNA based signatures and fulfill the criteria for a non-sputum triage test. The quality of the manuscript is provided by controlled experimental

procedures and by a clear exposure of the results. Some minor changes could improve the overall quality of the manuscript.

Thank you for your appreciation of our work. We have addressed all your comments and suggestions. In addition, to address comments from other reviewers, we have include extensive new data from new experiments in this revision (see also our replies to the comments of the other two reviewers).

Main concerns:

1. In the method section it is written that samples of the END TB cohort have been collected using Streck cell-free DNA collection tubes. I wonder why specific tubes for RNA preservation have not been used instead of those for DNA.

The samples were collected in the scope of another study, which aimed to evaluate the utility of circulating cell-free DNA in plasma. While we obtain good results regardless of collection tube (Streck cell-free DNA or K2EDTA), a future study evaluating the impact of RNA preservatives on the diagnostic performance may find improved performance.

2. In the methods section, cfRNA isolation paragraph, a range of volumes of plasma has been reported for the RNA isolation, does this mean that RNA has been isolated from different volumes of plasma in different samples? If so, authors should explain why. Could this differences have an impact on the quantification RNAs detected?

RNA was isolated from different volumes of plasma in different samples due to the limited volume of plasma we received from clinicians. We tested whether there is a relation between the input volume and the TB score, and did not observe a correlation.

3. At page 7, it is not clear how the Authors suggest to complement the chest X-ray results with the analysis of the 9-gene signature. Authors could explain better this aspect in the discussion section.

This follows on our remarks to reviewer #1: Whether plasma cfRNA profiling provides complementary information should be evaluated in future studies (ours was not designed to compare the accuracy of the two methods). For example, future studies may assess correlation between genes or gene signatures and CXR measurements (e.g. cavitation, opacity, texture, and shape). However, a major challenge is that CXR equipment is not routinely available in high burden countries. Our hope is that a plasma cfRNA signature can be more easily translated into a widely available, point-of-care test.

4. The specification of some acronyms, such as those of the algorithms used, could improve the reading of the results.

Thank you for this feedback, in the revised text we have done our best to make this more clear, specifically in the methods section.

5. At page 5, line 157, minimal and optimal sensitivities may have been inverted.

We have corrected the minimal and optimal sensitivities accordingly.

6. At page 6, a reference about the thresholds used in the Xpert method is appropriate.

We have added Reference 16 which describes good correlation between the Xpert thresholds, sputum positivity, and bacterial load.

REVIEWER COMMENTS

Reviewer #1 (Remarks to the Author):

no comments

Reviewer #1 (Remarks on code availability):

The authors have effectively addressed the raised queries. The manuscript is now suitable for acceptance and publication.

Reviewer #2 (Remarks to the Author):

The authors improved an already good manuscript and addressed the critical concerns. I look forward to seeing this published.

Reviewer #3 (Remarks to the Author):

The authors have replied to my main concerns about the article, except for point 2. If the analysis of cfRNA has been performed starting from different amounts of plasma input (where the lower input is less than a half of the higher one), how can authors state that the quantification of the cfRNA has not been affected? I think that the results should be normalized for the volume of input material to assess if the identified gene signature maintains its diagnostic potential. In my opinion, the author's response to point 2 is not convincing and this issue may fundamentally affect the quality to the manuscript

Point-by-point address of the specific comments raised by the editors and reviewers.

Reviewer #1 (Remarks to the Author):

no comments

Thank you for the appreciation of our work, and your many great suggestions during the review process.

Reviewer #2 (Remarks to the Author):

The authors improved an already good manuscript and addressed the critical concerns. I look forward to seeing this published.

We look forward to seeing this work published as well. Your feedback was very valuable, thank you for your time and effort.

Reviewer #3 (Remarks to the Author):

The authors have replied to my main concerns about the article, except for point 2. If the analysis of cfRNA has been performed starting from different amounts of plasma input (where the lower input is less than a half of the higher one), how can authors state that the quantification of the cfRNA has not been affected? I think that the results should be normalized for the volume of input material to assess if the identified gene signature maintains its diagnostic potential. In my opinion, the author's response to point 2 is not convincing and this issue may fundamentally affect the quality to the manuscript.

Thank you for the question. The signatures we report are based on the relative abundance of transcripts of specific genes: we normalize the counts of specific gene transcripts to the total number of gene transcripts measured. Because the gene signatures are based on a relative abundance measurement, and not the absolute abundance of transcripts isolated from plasma, it makes sense that the assay results are not dependent on the volume of plasma used. This is generally true for ensemble-level, bulk RNA-sequencing assays, which report fragments detected per million fragments measured, and for which the sample volume (tissue/cells/fluid volume) is not considered in differential gene expression analysis.

That said, in the scope of our experiments we have meticulously kept track of the plasma volumes used, and this allowed us to verify, with data, that the cfRNA signature score is indeed not affected by sample volume (see Figure below, Pearson's R: -0.036, p=0.57).). We include this analysis as supplementary figure S2 in the new version of the paper. We also show that although the range of plasma volumes used was 100 μ L-1000 μ L, most samples had a volume of 500 μ L -1000 μ L (n = 234) and few samples had a volume less than 500 μ L (n = 17, Figure below).

We thank you for your comments and suggestions.

Figure S2. No relation between plasma volume and signature score. A) Histogram showing the distribution of plasma sample volumes used in this study. **B)** Lack of correlation of the 6-gene TB score with plasma sample volume (Pearson correlation $r = -0.036$, $p = 0.57$). Color indicates disease status (pink = TB positive; blue = TB negative).

REVIEWERS' COMMENTS

Reviewer #3 (Remarks to the Author):

I have considered the new authors' reply to my concern regarding the evaluation of mRNA profiles by different starting amounts of plasma and the clarification of the Authors justify that the cfRNA signature score is not affected by sample volume. I agree now with the publication of the manuscript

Point-by-point address of the specific comments raised by the editors and reviewers.

Reviewer #3 (Remarks to the Author):

I have considered the new authors' reply to my concern regarding the evaluation of mRNA profiles by different starting amounts of plasma and the clarification of the Authors justify that the cfRNA signature score is not affected by sample volume. I agree now with the publication of the manuscript.

Thank you for the review and appreciation of our work.